# Utilization of Physiologically Based Pharmacokinetic Modeling in Pharmacokinetic Study of Natural Medicine: An Overview

**DOI:** 10.3390/molecules27248670

**Published:** 2022-12-08

**Authors:** Qiuyu Jia, Qingfeng He, Li Yao, Min Li, Jiaying Lin, Zhijia Tang, Xiao Zhu, Xiaoqiang Xiang

**Affiliations:** Department of Clinical Pharmacy and Pharmacy Administration, School of Pharmacy, Fudan University, Shanghai 201203, China

**Keywords:** physiologically based pharmacokinetic modeling, natural medicine, drug–drug interaction, gene polymorphism, special populations, new drug research and development

## Abstract

Natural medicine has been widely used for clinical treatment and health care in many countries and regions. Additionally, extracting active ingredients from traditional Chinese medicine and other natural plants, defining their chemical structure and pharmacological effects, and screening potential druggable candidates are also uprising directions in new drug research and development. Physiologically based pharmacokinetic (PBPK) modeling is a mathematical modeling technique that simulates the absorption, distribution, metabolism, and elimination of drugs in various tissues and organs in vivo based on physiological and anatomical characteristics and physicochemical properties. PBPK modeling in drug research and development has gradually been recognized by regulatory authorities in recent years, including the U.S. Food and Drug Administration. This review summarizes the general situation and shortcomings of the current research on the pharmacokinetics of natural medicine and introduces the concept and the advantages of the PBPK model in the study of pharmacokinetics of natural medicine. Finally, the pharmacokinetic studies of natural medicine using the PBPK models are summed up, followed by discussions on the applications of PBPK modeling to the enzyme-mediated pharmacokinetic changes, special populations, new drug research and development, and new indication adding for natural medicine. This paper aims to provide a novel strategy for the preclinical research and clinical use of natural medicine.

## 1. Introduction

Natural medicine, especially botanical drugs, is widely used in the fields of clinical treatment and health maintenance in many regions of the world, including China, India, Sumer, Egypt, and Greece [1]. In addition, extracting active ingredients from traditional Chinese medicine (TCM) and other natural plants, defining their chemical structure and pharmacological effects, and screening potential druggable candidates are also uprising directions in new drug research and development [2,3]. The study of the pharmacokinetics of natural plant components is a crucial step for the rational clinical use and further development of these drugs. The physiologically based pharmacokinetic (PBPK) model has been widely used in drug development for chemicals and has gradually been recognized by regulatory authorities in recent years, including the U.S. Food and Drug Administration (FDA) [4,5]. When it comes to natural medicine, the PBPK modeling can compensate for the shortcomings of traditional research methods, with unique advantages in speculating drug concentrations in target organs, interspecies extrapolation, pharmacokinetic study of special populations, and other aspects. Moreover, this computational approach is of particular significance to speed up the process and reduce the cost. Therefore, this review summarizes the applications of PBPK modeling in the research and development of natural medicine in order to promote the understanding of such mechanistic modeling and provide new tools and ideas for preclinical development and further clinical use of active components of natural medicine.

## 2. Overview of Pharmacokinetic Study of Natural Medicine Components

Natural medicine refers to the natural products and preparations from natural animals, plants, and microorganisms, with a particular pharmacological activity on the human body under the guidance of modern medical theory. Capsules, tablets, injections, liniments, and creams prepared from plant extracts are widely used in clinical practice. With the development and utilization of TCM and plant drug resources, especially after the great success of artemisinin, the research on the internal components of natural plants has become an important direction of global new drug investigations. Thanks to advancing analytical techniques and sensitive and accurate detection instruments, it is possible to determine the contents of active but trace components in plants, which significantly promotes the exploration of natural medicines. In recent decades, the pharmacokinetic studies of natural medicines have mainly focused on the pharmacokinetics of the main active components of TCM, the interactions between Chinese and Western medicine, the compatibility of TCM, and the effects of drug-metabolizing enzymes on the active ingredients in natural medicines [6,7,8,9,10,11,12,13].

Most of the published studies on the pharmacokinetics of natural drugs were based on healthy animals, or animal models under diseased states, to simulate the pharmacokinetics of natural drugs in vivo. Some studies directly measured plasma concentrations and obtained the concentration–time curve of the drug in humans. Taking triptolide as an example, Guo et al. [14] found that triptolide showed nonlinear elimination in rats. Li et al. [15] determined the serum content of triptolide in patients with rheumatoid arthritis. In vivo pharmacokinetic models of the active components of natural medicine were constructed individually in these studies, with some limitations. First, the animal or human plasma concentration obtained in the study could not represent the drug exposure in the target organ or tissue. Thus, a novel method is still strongly needed to obtain the drug level in those areas. Second, the conclusions of the studies cannot directly guarantee the safe use of single-plant drugs. Different from chemical drugs, the components of plant drugs are complex, and the active components are not ultimately revealed. A single component cannot represent a single plant, and the efficacy of plant drugs may be the comprehensive effect of multi-components and multitargets. Therefore, a method is needed to integrate the isolated results of pharmacokinetic studies of multiple components. Third, the number of patients who usually participate in determining plasma concentration is relatively small, resulting in significant bias and poor representation. At the same time, the participant differences in physiological status can lead to an extensive fluctuation range of drug plasma concentrations. Fourth, animal experiments are accompanied by long operational periods and species differences, making it challenging to apply the results directly to the human body. Fifth, due to the limitations of ethics and other aspects, it is not easy to conduct the pharmacokinetics study of natural drugs via clinical trials. Suppose there is a validated and reliable pharmacokinetic model which is easy to operate and can be directly applied to the human body. In that case, it can significantly reduce animal experiments, assist in understanding the disposition process of the drug in vivo, and significantly speed up drug research and development.

## 3. PBPK Modeling and Its Advantages in Pharmacokinetic Studies of Natural Medicine

### 3.1. Overview of the PBPK Modeling

A PBPK model is a mathematical model that simulates the absorption, distribution, metabolism, and excretion of drugs in various tissues and organs based on anatomical parameters and the physical and chemical properties of drugs [16,17,18]. The advantage of the PBPK model lies in using mathematical models to simulate the pharmacokinetic profiles of drugs in vivo, which can possibly partially replace animal experiments or clinical trials [19]. The PBPK model consists of two parts—body system and drug characteristics. The body system integrates the physiological and pathological conditions of the human body or other species, including blood perfusion rate, tissue, organ volumes, etc. [20]. The drug properties cover the physicochemical properties and in vitro absorption, distribution, and metabolic parameters of drugs, such as the inherent clearance of metabolic enzymes, plasma-protein binding rate, and membrane permeability. The PBPK model describes and combines the two parts, using a “bottom-up” method, that is, it predicts the dynamic process of drug exposure in vivo according to the body system parameters and drug parameters in vitro [21,22]. The PBPK model divides the body into several compartments according to anatomical and physiological structure, which represent each tissue and organ, and connect these compartments into a closed-model structure through blood circulation. According to the mass-balance differential equations, the inflow and outflow of drugs in each compartment are described, and the calculations are carried out by computer programs. Through simulation, the PBPK model can provide concentration–time curves of drugs and their metabolites in plasma and specific tissues and organs, which has great advantages in predicting bioavailability and in understanding the kinetic process of drug metabolism in vivo.

At present, the PBPK model has been widely used in the research of drug–drug interactions (DDIs) [23,24], pharmacokinetics of special populations [25], and pharmacokinetic predictions to calculate first-in-human doses [26,27]. A variety of applications have been accepted by the FDA in drug development and regulatory decisions [4].

### 3.2. Advantages of PBPK Modeling in Pharmacokinetic Studies of Natural Medicine

For the pharmacokinetic study of natural medicine, PBPK modeling and simulation can be a novel and efficacious approach. The PBPK model has unique advantages in the safety evaluation of natural drugs, DDIs, pharmacokinetic study of special populations, and development of new drugs with active components of natural medicines. Figure 1 shows the strategies for studying the pharmacokinetics of natural medicine using the PBPK model.

#### 3.2.1. The Drug Exposure of Natural Medicine Components in Target Organs and Tissues Can Be Studied by PBPK Modeling

In developing new drugs or clinical use, the toxicity and safety of natural drugs often attract great attention. The PBPK model is based on human anatomy; compartments represent organs or tissues, and the quantities of drugs flowing into and out of each chamber follow the mass balance. Therefore, the PBPK model can predict the drug exposure of toxic natural medicine components in target organs or tissues, clarify the dose–effect relationship, and provide a safe dose reference for the appropriate use of harmful natural medicines. At the same time, the PBPK model simulates the distribution of drugs and metabolites in various organs and describes the pharmacokinetic process of natural medicine active components and their metabolites, which can help in exploring the metabolic mechanism of active components or their metabolites.

#### 3.2.2. The PBPK Model Is an Effective Tool for Studying DDIs

Natural products have a variety of components and are often used in combination with other herbs or chemical drugs. DDIs may occur internally between the active components of herbs, or externally between herbs and chemical drugs. For example, St. John’s wort is an inducer of cytochrome P450 (CYP) enzymes (most importantly CYP3A4) and P-glycoprotein, which can alter the pharmacokinetics of drugs such as digoxin, tacrolimus, indinavir, warfarin, alprazolam, and simvastatin [28]. PBPK modeling and simulation have been advocated for the quantitative prediction of DDIs [29]. It has been successfully used to describe several potential DDIs, including silibinin with raloxifene [30], glycyrrhizin with rifampin [31], curcumin with imatinib, and bosutinib [32]. The PBPK models of many active components of natural medicines can be constructed to mimic enzymatic interactions simultaneously [33]. This is suitable for the pharmacokinetic study of natural medicines characterized by multi-components.

#### 3.2.3. The PBPK Model Has Unique Advantages for Solving the Problem of Safe Use of Natural Medicine in Special Populations

Special populations, such as children, pregnant women, the elderly, and patients with renal or hepatic impairment, often face more significant risks when using natural medicine. Many clinical studies in special populations cannot be carried out because of ethical restrictions. The PBPK model has robust extrapolation and prediction functions, which can solve this problem. First, the PBPK model for healthy people will be established. Then, the model will be extrapolated to the particular population by adjusting the physiological parameters (such as protein binding rate, enzyme/transporter-related parameters, etc.) in the specific physiological or pathological state so that the pharmacokinetic behaviors of the active components of natural drugs in special populations can be predicted. It can reference dose adjustments and the safe use of natural medicine for special populations.

#### 3.2.4. The PBPK Model Has More Potential in the Research and Development of New Drugs Based on Natural Medicine

As a part of new drug development, finding potential monomer components from natural medicine, exploring the pharmacokinetic processes of monomer components, and integrating the pharmacokinetic information of these monomer components to establish a PBPK model helps in understanding the human pharmacokinetic characteristics of active components. With a strong extrapolation ability, the PBPK model can realize cross-species extrapolation from animal to human and solve the ethical difficulties. Based on that, the first dose in humans could be determined to decrease the risk of trial failure, reduce animal experiments, and improve the speed and efficiency of new drug research and development.

## 4. Applications of the PBPK Model in the Study of Pharmacokinetics of Natural Medicine

### 4.1. The Enzyme-Mediated Pharmacokinetic Changes in Natural Drugs

Gene polymorphism is significant in clinical practice. Genetic polymorphisms in a population may act to predispose individuals to adverse reactions or treatment failure [34]. Moreover, the use of natural medicine products continues to rise, partly because of the widespread misconception that “natural” is synonymous with “safety”. Patients often use natural medicine products to try to alleviate the disease or to supplement the treatment plan without professional consultation. As a result, natural medicines are usually taken in combination with prescription and over-the-counter drugs, likely triggering drug interactions that risk changing the pharmacodynamics and efficacy. Both gene polymorphisms and drug interactions lead to changes in the abundance of enzymes in the human body, resulting in changes in the pharmacokinetics of drugs in the population due to different metabolic rates, which will affect drug exposure and cause concerns about drug safety and efficacy. PBPK modeling and simulation can quantitatively predict the effects of gene polymorphism on the pharmacokinetics of natural drugs and study the DDIs between natural pharmaceutical preparations and chemical drugs.

#### 4.1.1. The effects of Gene Polymorphism on the Pharmacokinetics of Natural Drugs

Cannabis has been used to treat diseases and for recreational purposes for centuries [35], and it is important to determine its metabolism process in the human body. Wolowich et al. [36] studied the pharmacokinetics of an active component of cannabis delta-9-tetrahydrocannabinol (THC) and its main metabolites, 11-hydroxy-delta-9-tetrahydrocannabinol (11-OH-THC) and 11-nor-9-carboxy-delta-9-tetrahydrocannabinol (THC-COOH) in healthy volunteers with known CYP2C9*3 status through minimal PBPK modeling, noncompartmental analysis, and compartmental modeling. This study partially revealed the complexity of cannabis disposition in the human body. It was found that CYP2C9*3 polymorphism and the hepatic diffusional barrier contributed to the variability in THC disposition. The CYP2C9*3 phenotype reduced the exposure of THC-COOH because THC production decreased in this polymorphism. The activity of THC-COOH has been reported [37], implying that the exposure difference in individuals homozygous for CYP2C9*3 may become therapeutically relevant, indicating that dose adjustments should be considered in this subpopulation [36].

The impact of CYP2B6 polymorphisms on the interactions of efavirenz with lumefantrine is used to treat malaria in children. The exposure of lumefantrine in patients could be reduced when combined with antiretroviral efavirenz, which may lead to increased malaria recrudescence rates and treatment failure. Zakaria used the PBPK model to evaluate efavirenz’s impact in reducing lumefantrine’s pharmacokinetics by CYP3A4 induction in African pediatric populations. The model also considered the impact of the CYP2B6 polymorphism in DDIs between efavirenz and lumefantrine. The study indicated that lumefantrine concentrations on day seven were significantly lower in the population of the *6/*6 CYP2B6 phenotype (*p* < 0.001) compared to the population of *1/*1. Therefore, it is recommended that the artemether–lumefantrine treatment regimen be extended from 3 days to 7 days in *6/*6 genotype groups [38].

The herbal hepatoprotective drug, silybin A, inhibits CYP2C9 and CYP3A4 enzymes. Thus, it may interact with losartan, the substrate of CYP2C9 and CYP3A4. Tanveer et al. established a PBPK model to simulate the pharmacokinetics of losartan with or without coadministration of silybin A in populations of different CYP2C9 genotypes to study DDIs. It was found that silybin A was a weak CYP inhibitor of losartan in the population of CYP2C9*1/*1 and CYP2C9*1/*3, while there was a moderate pharmacokinetic interaction between silybin A and losartan in CYP2C9*1/*2. The results implied that clinical strategies (losartan dose and administration frequency) could be adjusted according to individual genotypes. CYP2C9*1/*1 and CYP2C9*1/*3 genotypes may not need losartan dose adjustments, but dose adjustments may be considered in CYP2C9*1/*2 genotypes [39].

#### 4.1.2. Study the Drug Interaction with Natural Medicine in Oncology Using PBPK Modeling

Many cancer patients take herbal drugs in addition to prescription drugs. Venkatesh et al. established a PBPK model to predict the extent of interactions between constituents in herbs and food, including grapefruit juice (bergamottin), turmeric (curcumin), and St. John’s grass (hyperforin), with oncology drugs (acalabrutinib, osimertinib, or olaparib). The corresponding interaction mechanisms (CYP3A, P-glycoprotein, and breast-cancer resistance protein), in vitro parameters, and clinical data were incorporated into the PBPK model. Prospective simulations with grapefruit juice and turmeric showed clinically minor-to-insignificant changes in exposure (<1.50-fold) to acalabrutinib, osimertinib, and olaparib. However, the interaction risk between acalabrutinib and curcumin was increased by 1.57 times. Taking St. John’s wort with acalabrutinib, Osimertinib, or olaparib has a moderate risk (ratio between 1.5 and 2) of DDIs [40]. This quantitative clinical-pharmacology-modeling framework will simplify the safety assessments of herbal products, assist with the management of DDIs, and ultimately facilitate the safe clinical use of oncology drugs.

In addition to the several aforementioned tumor drugs, DDIs between natural medicines and imatinib and bosutinib have been widely investigated. First, PBPK modeling can predict the effects of curcumin on the metabolism of imatinib and bosutinib by PBPK modeling. Curcumin is a natural active component isolated from turmeric with anticancer and hepatoprotective effects [41]. It has been identified as an effective inhibitor of the CYP3A enzyme and the ABCB1 and ABCG2 transporters. However, little is known about the potential interactions between curcumin and anticancer drugs, especially when different concentrations and new formulations are used [42]. The potential interactions could significantly impact their total body exposure (bioavailability). To predict the possible effects of curcumin coadministration on systemic exposures of imatinib and bosutinib, Adiwidjaja et al. established a PBPK model for curcumin formulated as solid lipid nanoparticles (SLN). Curcumin had potent, reversible inhibition of CYP3A4-mediated N-demethylation of imatinib and bosutinib and CYP2C8-mediated metabolism of imatinib in vitro. However, PBPK model simulations predicted only a 10% increase in systemic exposure to imatinib and bosutinib under the coadministration of SLN curcumin at the recommended dosing regimen (320 mg twice daily for 14 days), which is unlikely to be of clinical importance. PBPK simulations with different doses of curcumin showed that the interaction with imatinib and bosutinib would have clinical significance when more SLN curcumin was used (at least 3.2 g and 1.6 g, respectively) [32].

Moreover, the potential for pharmacokinetic interactions between Schisandra sphenanthera and bosutinib, but not imatinib, has recently been reported. The clinical therapeutic effect of Schisandra sphenanthera and its natural products is hepatoprotective [33]. It is known that patients treated with tyrosine kinase inhibitors take complementary medicines simultaneously, including Schisandra sphenanthera, to help reduce adverse effects [43]. Several bioactive lignans, namely schisantherin A (STA), schisandrin A (SIA), and schisandrol B, have been identified as being related to CYP3A enzymes. To investigate the potential interactions between *Schisandra sphenanthera* and imatinib or bosutinib, the PBPK models for interactions between Schisandra lignans and midazolam and tacrolimus were successively established and verified based on existing clinical pharmacokinetic and herb-drug interaction data. Then, the PBPK models for interactions between Schisandra lignans, imatinib, and bosutinib were developed. As result, it was predicted that the interaction between imatinib and Schisandra lignans was unlikely to be clinically significant. Conversely, bosutinib systemic exposure was expected to triple when *S. sphenanthera* was given at a clinically relevant dose [33].

In another study, combined with the in vitro experimental data and the published clinical pharmacokinetic data, the PBPK models were established to predict natural product–drug interactions between goldenseal, berberine, imatinib, and bosutinib. These models explain the reversible and irreversible (mechanism-based) inhibition of CYP3A enzymes and the inhibition of P-glycoprotein transporters. There is a potentially significant clinical interaction between goldenseal extract and bosutinib, but not imatinib. Thus, dose adjustment should be considered when coadministration is needed for treatment [44].

In clinical practice, the Wuzhi capsule can reduce cyclophosphamide toxicity in the kidneys and liver [45]. To study the effects of the main constituents of the Wuzhi capsule, SIA, and STA, on the pharmacokinetics of cyclophosphamide, Chen et al. developed reliable PBPK models of SIA, STA, and cyclophosphamide to predict the DDIs caused by SIA and STA. The results showed that the area under the plasma concentration–time curve (AUC) of cyclophosphamide was increased by 18% and 1% when administered in combination with STA and SIA at a single dose, respectively, and increased by 301% and 29% when administered in combination with STA and SIA at multiple doses, respectively. The maximum concentration of cyclophosphamide was increased by 75% and 7% when combined with STA and SIA at multiple doses, respectively. This study showed that STA had a more significant inhibitory effect on cyclophosphamide metabolism than SIA, and cyclophosphamide’s adverse drug reactions and toxicity should be closely monitored. Dosing adjustment of cyclophosphamide could be considered when coadministered with the Wuzhi capsule [46].

#### 4.1.3. Study of the Drug Interactions with Natural Medicine in Immunosuppression by PBPK Modeling

The Wuzhi capsule is often prescribed with tacrolimus in China to alleviate drug-induced hepatotoxicity. Considering DDIs and gene polymorphisms, the effects of several active components on the pharmacokinetics of tacrolimus have been studied. As the main active components of the Wuzhi capsule, STA and SIA can inhibit the CYP3A enzymes and increase the systematic exposure of tacrolimus. Zhang et al. quantitatively analyzed the contributions of STA and SIA in the Wuzhi capsule on the pharmacokinetic profile of tacrolimus based on PBPK modeling [47]. Moreover, CYP3A5 is an important metabolic enzyme of tacrolimus. Approximately 40–50% of the variability in tacrolimus dose requirement is due to the polymorphism of CYP3A5 [48]. Another study [49] showed that STA had a time-dependent and reversible inhibition effect on CYP3A4, but only a reversible inhibition effect on CYP3A5, according to in vitro experiments. Nevertheless, SIA had time-dependent inhibition on CYP3A4 and 3A5 and reversible inhibition on CYP3A5. To predict the impact of CYP3A5 polymorphisms on DDIs between tacrolimus and schisantherin A/schisandrin A, a PBPK model was established with the CYP3A5 polymorphism included. The results showed that tacrolimus exposure was increased 2.70- and 2.41-fold in CYP3A5 expressers and nonexpressers, respectively, after the multidose simulations of STA. SIA also increased tacrolimus exposure, but to a lesser extent than STA.

Moreover, Fructus schizandrol A (SZA) and schizandrol B (SZB) are two active components of the Wuzhi capsule. Metabolic study results showed that SZB shows both reversible and time-dependent inhibition on CYP3A4 and CYP3A5, while SZA shows time-dependent inhibition to a limited extent through CYP3A4. In addition, the PBPK model predicted that multidose SZB would increase tacrolimus exposure by 26% and 57% in CYP3A5 expressive and nonexpressive individuals, respectively. The results can provide reference and help for patients with different CYP3A5 genotypes in long-term DDIs between tacrolimus and Wuzhi capsules. Particular attention should be paid to the coadministration of the Wuzhi capsule and tacrolimus [50].

Cyclosporine A is also commonly used to alleviate rejection after organ transplantation. It has been reported that Wuzhi capsules could change the pharmacokinetics of cyclosporine A [51]. The AUC of cyclosporine A, in combination with STA, increased by 47% and 226%, respectively, and in combination with SIA, increased by 8% and 36%, respectively. The PBPK model fully described the pharmacokinetics of cyclosporine A, SIA, and STA. Compared with SIA, STA has a more significant inhibitory effect on cyclosporine A metabolism. Our results show that the dose of cyclosporine A can be reduced to maintain a similar profile compatible with the Wuzhi capsule.

### 4.2. Pharmacokinetics of Natural Medicine Active Components in Special Populations

It is often difficult for special populations, including children, pregnant women, the elderly, and patients with liver or kidney insufficiency, to determine the safe dose in clinical practice. For instance, the risk of a drug overdose in patients with liver and kidney injury is significantly increased due to fluctuating physiologic status. In addition, clinical trials in special groups are often waived due to ethical problems. Thus, it is hard to study the pharmacokinetics of natural medicine active components in vivo in special populations. PBPK modeling can tackle this problem by predicting the pharmacokinetic process of compounds in vivo based on physiological and pathological changes.

#### 4.2.1. THC/11-OH-THC PBPK Modeling in Pregnant Women

Continuing use of cannabis during pregnancy is associated with poorer neonatal outcomes [52] and neurodevelopmental consequences [53]. THC and its active metabolite, 11-OH-THC, are the main psychoactive constituents of cannabis. Prospective clinical studies to measure maternal and fetal drug exposure are not ethically feasible but can be simulated using the PBPK model. Patilea-Vrana et al. developed and validated a linked PBPK model of THC and 11-OH-THC in a healthy, nonpregnant population after intravenous and inhalation administration, and then extrapolated to pregnant women by taking into account the gestational physiological changes, such as changes in drug-metabolizing enzyme activity and drug-binding protein expression. Simulation results showed that the THC plasma AUC did not change during pregnancy, but 11-OH-THC plasma AUC decreased by up to 41%. The linked THC/11-OH-THC PBPK model can be extrapolated and then predict exposure in special populations, DDIs, or the impact of gene polymorphism in the future [54].

#### 4.2.2. Evaluating the Safety of Tripterygium Wilfordii in Patients with Liver Injury by PBPK Modeling

In clinical practice, Tripterygium wilfordii treats autoimmune diseases such as rheumatoid arthritis and systemic lupus erythematosus. Triptolide is the main active ingredient with immunosuppressive and anti-inflammatory effects, but also the main component causing hepatotoxicity. To evaluate the safety of Tripterygium wilfordii products in patients with liver injury, Wu established a rat PBPK model of triptolide and then extrapolated it to a human model. The pharmacokinetics of triptolide in healthy and liver injury populations was predicted by the PBPK model. The results showed that the metabolic rate of triptolide decreased and drug exposure was significantly elevated in patients with moderate and severe liver cirrhosis, which may aggravate the hepatotoxicity of triptolide in such patients and cause serious safety problems [55]. Therefore, when Tripterygium wilfordii preparation is used, more attention should be paid to the liver function of patients and dosage adjustment in time to ensure safety.

### 4.3. PBPK Modeling Can Help Speed Up the Research and Development of New Natural Medicines and Add New Indications

There has been an increasing demand to predict human pharmacokinetics as early as possible to help select the best candidates for further development and avoid drug discovery failures due to poor pharmacokinetics. The PBPK model can predict the concentration–time profiles of compounds in plasma and human target organs or tissues, giving an idea of such compound performances, which makes it easy to understand the pharmacokinetic properties of novel compounds. Thus, in the drug screening stage, the PBPK model can assist in obtaining drug information. When a new drug is about to enter a clinical trial, the PBPK model can assist in choosing the first dose in humans to speed up the process. At the same time, for existing natural compounds, PBPK modeling can be used to predict the concentration in new target tissues and increase new indications.

#### 4.3.1. Dosing Determination for Natural Medicine Using PBPK Modeling

In order to achieve optimal ratios of red-clover components (formononetin, biochanin, A. daidzein, and genistein) and obtain optimal efficacy, a PBPK/PD model was successfully established to describe the time course of concentration and effect. The red-clover bioactive substances and their metabolites in different animal species, including humans, were included, along with the enterohepatic cycling process. The pharmacodynamic model described the rate of bone formation and resorption. According to the simulation results, a recommended daily dosage of 5 to 200 mg was recommended [56].

Andrographolide has a potent antiviral effect in the treatment of COVID-19. Talapphetsakun established a perfusion-limited PBPK model in mice and then extrapolated it to rats, dogs, and humans. According to the results, there is absorption saturation of the high dose (12 g) of andrographolide. Based on simulation results, it was proposed that an oral dose of 200 mg every 8 h can keep the trough level of andrographolide in the lungs above the reported anti-SARS-CoV-2 half-maximal inhibitory concentration, but lower than the pulmonary toxicity threshold [57].

#### 4.3.2. Local Exposure Estimation of Natural Medicine Using PBPK Modeling

Rivero-Segura et al. screened 100 compounds in silico isolated from the most commonly used Mexican herbal medicine. Only three compounds could target the critical proteins of SARS-CoV-2 stably, while only one (cichoriin) was safe, based on the docking results and toxicological properties. Cichoriin has a strong affinity to the main targets of SARS-CoV-2. To determine whether cichoriin could achieve sufficient exposure locally against COVID-19, PBPK models were constructed using data from other coumarins with similar structures. The simulation showed that when injected intravenously at 100 mg/kg, the concentration of cichoriin might reach acceptable levels in plasma, intracellularly, and the highest concentration in the lung. The PBPK model makes it easier to understand the pharmacokinetics properties of novel compounds and obtain information about their local exposure to the target organs and tissues [58].

Mitragynine is the major psychoactive alkaloid in kratom leaves and acts as a μ opioid agonist with analgesic, antidepressant, and other pharmacological effects. A PBPK model was successfully established by incorporating important biologically related features, including breast-cancer resistance protein in the brain, CYP3A4-mediated liver metabolism, and diffusion-limited transport in fat. The model can predict the blood and brain concentration–time curves of mitragynine in rats and humans under different administration scenarios and can guide the formulation of better local penetration [59].

Deoxypodophyllotoxin (DPT) is a phytochemical with antitumor pharmacological activity. A PBPK model including tumor compartment was established to predict DPT exposure in plasma, tumor tissue, and main normal tissues of NCI-H460 tumor-bearing mice. Then PBPK model was linked with a PD model based on in vitro cytotoxicity test results, and the PBPK-PD model successfully predicted the tumor growth of NCI-H460 tumor-bearing mice during multidose DPT treatment. The results showed that the antitumor effect in vivo could be predicted based on in vitro cytotoxicity test by the PBPK-PD model. This method can facilitate and accelerate the screening of anticancer drug candidates and the design of drug delivery regimens in drug discovery [60].

#### 4.3.3. Extrapolation to Humans Pharmacokinetically and Pharmacodynamically Using PBPK Modeling

Soraphen A is a natural macrolide product produced by the myxobacterium Sorangium cellulosum. It is effective against the dengue virus. A mouse PBPK/PD model for dengue virus was established and extrapolated to humans. The clinical situation predicted a reduction in viremia by more than one log_10_ unit for an intravenous infusion regimen of Soraphen A [61].

To assess the PKs and PD of 5-methoxypsoralen that is standardized in an approved drug in Brazil, a PBPK model was established in rats using in vitro and in vivo studies. Then, the model was extrapolated to humans to support psoralen and ultraviolet type A therapy. PBPK model prediction in humans supported a once-every-two-day regimen for optimal melanin production. The method can be suitable for supporting dose selection strategies and studying the effects of drugs on melanocyte recovery [62].

### 4.4. Other Applications of PBPK Modeling in the Pharmacokinetic Study of Natural Medicine

The effects of intestinal metabolism and enterohepatic circulation on bioavailability and systemic disposal of resveratrol in rats and humans could be investigated by PBPK modeling. Resveratrol is a natural polyphenol plant health hormone. The PBPK model simulation illustrated the significant contribution of intestinal first-pass metabolism to the systemic elimination of resveratrol and the differential effect of enterohepatic circulation on systemic exposure to resveratrol in rats and humans. After partial modification and verification, the PBPK model can optimize the drug delivery regimen and predict the interaction between resveratrol natural products and drugs in various clinical scenarios [63].

The risk of drug-induced liver injury poses a major challenge for developing natural products derived from TCMs. Li et al. developed a method integrating a support vector machine classifier and PBPK modeling. They predicted the dosing schedules of triptolide, emodin, matrine, and oxymatrine, corresponding to the different probabilities of hepatotoxicity. The safe dosing regimen of oxymatrine estimated by this method (367 mg thrice a day) was close to the clinically recommended regimen (200–300 mg thrice a day), which proved the reliability of the technique. The results showed that the technique could be used to predict the daily multidose administration regimen of the compound and provide a reference for the safety evaluation and development of natural products derived from TCMs [64].

PBPK modeling could be used for exposure, toxicity, and risk assessment of Pyrrolizidine alkaloids in food and phytomedicine. Petasites japonicus is consumed as a wild vegetable or used in folk medicine in Japan. Neopetasitenine is the major form of pyrrolizidine alkaloid in Petasites japonicus, which exists at high concentrations with its carcinogenic deacetylated metabolite, petasitenine. The plasma levels of neomymatrine and petasitenine were predicted by establishing a PBPK model in rats and extrapolating the model to humans with an allometric scaling method. PBPK simulations showed that daily Petasites japonicus will cause sustained dangerous concentrations of deacetylated petasitenine in human plasma or liver [65].

PBPK model is an excellent tool for studying the permeability of TCM and predict human absorption. Liu et al. established the PBPK model of several aglycones (quercetin, daidzein, formononetin, genistein, and glycyrrhetinic acid) in rats and beagle dogs. They extrapolated the model to humans by allometric scaling, followed by predictions of absorption fraction and bioavailability of aglycone components in humans. The predicted absorption fraction in humans showed a similar data range and trend, posing a good correlation with predicted results by the single-pass intestinal perfusion model [66].

The applications of the PBPK model in the study of the pharmacokinetics of natural medicine mentioned above are summarized in Table 1.

## 5. Discussion

As a reliable quantitative method for pharmacokinetic studies, PBPK modeling has been used increasingly in the research, development, and clinical administration of natural medicine. The PBPK model has unique value and significance for studying the pharmacokinetics of natural drugs. (i) The PBPK model can significantly reduce unnecessary trials and lower experimental costs. Traditional pharmacokinetic studies in vivo are generally based on animal experiments, resulting in a large amount of experimental animal and material consumption [67]. PBPK modeling can reduce the use of experimental animals and only requires in vitro data to build bottom-up modeling. In a recent study, the pharmacokinetics of seven commonly used microsomal enzyme probe substrates were well-predicted by PBPK modeling [68]. It was proved that PBPK modeling is a feasible strategy to practice the 3R principles and a valuable tool for pharmacokinetic research. (ii) PBPK modeling can predict the pharmacokinetics among different species and speed up the research process for the pharmacokinetic analysis. With advanced computational technology, PBPK modeling can make drug screening faster and identify clinically significant and exploitable compounds with good pharmacokinetic properties, greatly shortening the preclinical process and assisting in choosing the first dose in humans. (iii) When applied to guide clinical drug use, PBPK modeling can help achieve accurate dosing in different populations and reduce adverse drug reactions.

Natural drugs have been used worldwide for centuries, especially in developing countries. However, natural medicines are generally defined as dietary supplements, not drugs, and are not regulated by governmental health agencies, e.g., the FDA and the European Medicines Agency [69]. It has been reported that 50% of Americans take dietary supplements, mostly composed of natural botanical products [70]. Although natural medicines are widely used, the pharmacokinetics and pharmacodynamics of many natural medicines are not clear. In this review, the related research and application of the PBPK model in natural medicine are summarized for the first time. Compared with traditional animal experimental research, there are fewer articles to establish the PBPK model in silico to study the pharmacokinetics of natural medicine. However, the PBPK model includes the mechanisms that affect ADME (such as CYP450 enzymes, glucuronosyltransferases enzymes, transporters such as P-glycoprotein, and breast-cancer resistance proteins). This bottom-up research method cannot be achieved by traditional pharmacokinetic research. Recently, the uprising research ideas mainly focus on the metabolic and elimination mechanisms of natural drugs in vivo, DDI predictions, gene polymorphisms, concentration predictions in target organs or tissues for drug development or new indication adding, and drug dosage adjustments for special populations using PBPK models. Among them, DDI research accounts for a larger proportion because natural medicine is generally used as a supplement in combination with chemical drugs, while natural medicine plays a role as a perpetrator. As concluded in Table 1, the interaction could be so significant that it requires dosing adjustments based on the exposure change.

Despite the promising applications, it should be noted that PBPK modeling is not directly applicable to the pharmacokinetic study of natural medicine according to its modeling characteristics. Different from chemical drugs, natural-medicine PBPK modeling has more difficulties. Firstly, choosing a component as the main object and integrating remainings is a complex problem for natural medicine in PBPK modeling. Secondly, the lack of standardization of TCM products or other botanical drugs leads to significant differences in the content of components. Thus, as described in the PBPK modeling strategy of natural medicine in Figure 1, the dose of the key active component should be determined first. Thirdly, considering the multicomponent characteristics of natural drugs, it is difficult to ignore the influences of internal interactions—which are complex and require further research and discussion—on pharmacokinetic studies. In addition, the relevant pharmacokinetic data of natural drugs in clinical practice are less than that of chemical drugs. These factors restrict the application of the PBPK model in the pharmacokinetic study of natural medicine. Therefore, PBPK modeling is more suitable for studying monomer compounds with clear structures and principal components with high content in natural drugs to further explore the single and synergistic effects of multiple components in pharmacokinetics. Furthermore, PBPK modeling requires a large number of physiological parameters, such as the volume and blood flow rates of various tissues and organs, as well as related parameters of drug metabolism in vivo, including drug solubility, plasma-protein binding rate, enzyme and transporter parameters, etc. Obtaining these parameters requires intensive literature review and benchwork. In addition, as a mathematical model, PBPK modeling also requires developing computational algorithms to support complex calculations and more sophisticated frameworks. With the development of related technologies, PBPK modeling is expected to be more optimized and advanced enough for natural medicine.

## 6. Methods

PBPK studies from 2018 to 2022 that were reported to be related to natural medicine were researched based on the “Mesh terms OR Entry terms” strategy in four databases: PubMed, Web of Science, Embase and the China National Knowledge Infrastructure (CNKI). The relevant PBPK studies on natural medicine were identified using the following search terms: ((Pharmacokinetics)) AND ((Physiologically Based Pharmacokinetic) OR (PBPK) OR (Simcyp) OR (PK Sim) OR (Gastroplus) OR (MATLAB) OR (WinNonlin) OR (Berkeley Madonna) OR (R Software) OR (STELLA) OR (MCSim) OR (RVis) OR (acslX)) AND ((Dietary Supplements) OR (Plant Preparations) OR (Plants, Medicinal) OR (Herbal Medicine) OR (Phytochemicals) OR (Phytomedicine) OR (Herb) OR (Herba) OR (Herbal) OR (Herbal Preparations) OR (Herbaceous Agent) OR (Constituent) OR (Vegetable Medicine) OR (Vegetable Drug) OR (Natural Drug) OR (Natural Compound) OR (Natural Medicine) OR (Plants) OR (Crude Drugs) OR (Traditional Chinese Medicine) OR (Traditional Medicine) OR (Chinese Herbal Monomer) OR (Galenical) OR (Galenical Pharmaceuticals) OR (Galenical Extracts) OR (Galenical Preparations) OR (Botanical Drug) OR (Botanical Natural Product)). The inclusion criteria were: 1) pharmacokinetic studies based on PBPK modeling; 2) studies involving natural medicine. The exclusion criteria were surveys, case reports, and abstracts of congress meetings or conference proceedings. Finally, 24 typical studies were included and discussed in this review.

## 7. Conclusions

This review summarizes the applications of the PBPK model in the pharmacokinetic study of natural drugs, including DDI, gene polymorphism, special populations, and new drug research and development. Currently, the PBPK modeling of natural medicine focuses on the monomer components. With the development and advancement of PBPK modeling, it is expected that multi-components will be studied comprehensively in the future.

## Figures and Tables

**Figure 1 molecules-27-08670-f001:**
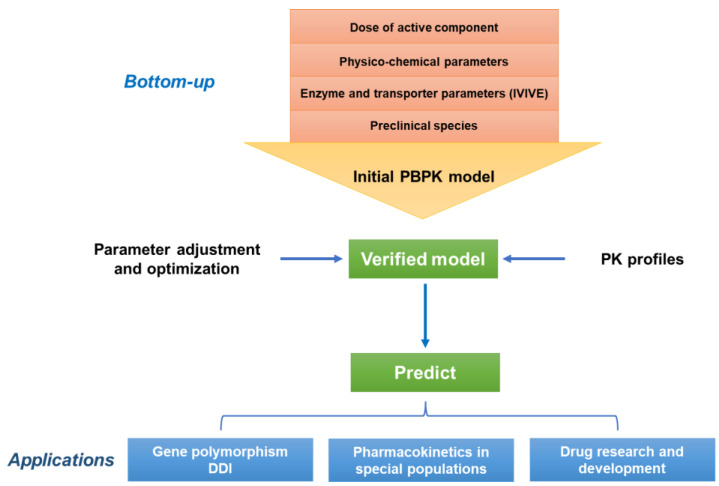
Strategies for studying the pharmacokinetics of natural medicine using the PBPK model. IVIVE: in vitro–in vivo extrapolation; PBPK: physiologically based pharmacokinetic; PK: pharmacokinetic; DDI: drug–drug interaction.

**Table 1 molecules-27-08670-t001:** Applying the PBPK model in the studies of the pharmacokinetics of natural medicine.

Applications		Examples		Regimen Adjustment	References
The enzyme-mediated pharmacokinetic changes in natural drugs	The effect of gene polymorphism on the pharmacokinetics of natural drugs	The special pharmacokinetics of THC in individuals homozygous for CYP2C9*3		NA	[36]
		The impact of CYP2B6 polymorphisms on the interactions between efavirenz and lu-mefantrine		Artemether and lumefantrine regi-men should be ex-tended from 3 days to 7 days in *6/*6 genotype groups	[38]
	The drug interaction with natural medicine in oncology	**Perpetrator drugs: natural medicine** **(active components)**	**Victim drug:** **chemical drugs**	**Regimen** **adjustment**	**References**
		Grapefruit juice (bergamottin); Turmeric (curcumin); St. John’s grass (hyperforin)	Acalabrutinib; Osimertinib; Olaparib	NA	[40]
		Turmeric (curcumin)	Imatinib; Bosutinib	NA	[32]
		Schisandra sphenanthera (SIA, STA and schisandrol B)	Imatinib; Bosutinib	Bosutinib should be reduced from 400 to 150–200 mg s after coadministration with Schisandra lignans	[33]
		Goldenseal; Berberine	Imatinib; Bosutinib	NA	[44]
		Wuzhi Capsule (SIA and STA)	Cyclophosphamide	NA	[46]
	The drug interaction with natural medicine in transplant	Wuzhi Capsule (SIA and STA)	Tacrolimus	NA	[49]
		Wuzhi Capsule (SZA and SZB)	Tacrolimus	NA	[50]
		Wuzhi Capsule (SIA and STA)	Cyclosporine A	NA	[51]
Pharmacokinetics of natural medicine active components in special populations		**Natural medicine** **(active components/metabolites)**	**Special populations**	**Regimen recommendation**	**References**
		Cannabis (THC and 11-OH-THC)	Pregnant women	NA	[54]
		Tripterygium wilfordii (triptolide)	Patients with liver injury	NA	[55]
Speed up the research and development of new drugs	Dosing determination for natural medicine using PBPK modeling	Research and development of a phytoestrogen product for the prevention or treatment of osteoporosis using red clover		5–200 mg daily dosage of total isoflavone	[56]
		Predict the concentration of andrographolide in the lungs to initially evaluate the potential efficacy of the proposed COVID-19 regimen		200 mg of andrographolide orally q8h	[57]
	Local exposure estimation of natural medicine	In-silico screening of natural medicine isolated from Mexican herbal medicines against COVID-19		Cichoriin IV at 100 mg/kg	[58]
		Predict the blood and brain concentration–time curves of mitragynine in rats and humans		NA	[59]
		Predict the PK of deoxypodophyllotoxin in mice to accelerate the screening of anticancer drug		NA	[60]
	Extrapolation to humans pharmacokinetically	The treatment of dengue infections applied to Soraphen A		NA	[61]
		PK and PD assessment of 5-methoxypsoralen in humans to support psoralen and ultraviolet type A therapy		Supported a once-every-two-day regimen for optimal melanin production	[62]
Other applications of PBPK modeling in the pharmacokinetic study of natural medicine		The effects of intestinal metabolism and enterohepatic circulation on bioavailability and systemic disposal of resveratrol in rats and humans		NA	[63]
		Prediction of oral hepatotoxic dose of natural products derived from TCM based on support vector machine classifier and PBPK modeling		regimen of oxymatrine estimated by this method (367 mg thrice a day) was close to the clinically recommended regimen (200–300 mg thrice a day)	[64]
		Exposure, toxicity and risk assessment of Pyrrolizidine alkaloids in food and phyto-medicine		NA	[65]
		Study the permeability of aglycones in TCM and predict human absorption		NA	[66]

PBPK: physiologically based pharmacokinetic; NA: not available; THC: delta-9-tetrahydrocannabinol; SIA: schisandrin A; STA: schisantherin A; SZA: schizandrol A; SZB: schizandrol B; 11-OH-THC: 11-hydroxy-delta-9-tetrahydrocannabinol; PK: pharmacokinetic; PD: pharmacodynamic; TCM: traditional Chinese medicine.

## Data Availability

Not applicable.

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
