# Peer review of "Utilization of Physiologically Based Pharmacokinetic Modeling in Pharmacokinetic Study of Natural Medicine: An Overview"

_molecules, 2022, doi:10.3390/molecules27248670_

Round 1
Reviewer 1 Report
The manuscript of Qiuyu Jia and Qingfeng He has been revised. The paper focused on the use of Pharmacokinetic modeling in pharmacokinetic study of natural drugs and their diverse applications, with special emphasis on drug-drug interactions, special populations, dosage administration and new drug research and development. Overall, the review is well-written and sounds interesting for the audience of Molecules. The topic proposed by the authors is extensive and varied, and the synthesis made by the authors is mostly laudable. In general, the bibliographic references are diverse and cover the current achievements in the topic. However, the ratio of references to the size of the review is somewhat poor (54 references for 10 pages of text) and more references should be included to support different sections, specially SARS CoV-2 and cancer (sections 4.3.2 and 4.1.2). The literature is diverse, encompassing a huge range of researchers in the area. I recommend major revision before the paper can be published in Molecules.
-Due to the current great interest, the authors should emphasize and expand their work in the SARS CoV-2 topic, including more reference of the possible applications and searching of new drug for SAR-CoV-2 treatment.
- A new paragraph or section should be included about the use of PBPK models in solving antibiotic resistance problem and obesity, the both health problems can be treated with natural medicaments.
-Table 1 must be implemented including data of drug dosage adjustment using PBPK models and the pertinent comments should be included in discussion section.
-A section with the main conclusions derived from the work should be included at the end of the manuscript.
Author Response
Please see the attachment "Response to reviewer #1.docx".

Reviewer 2 Report
PBPK modeling has been a valuable tool for regulatory science and can become a powerful tool in the hands of clinicians. The scarcity of user-friendly tools for modeling has limited the widespread utility of PBPK. Commercial PBPK software programs are well accepted for various PK analysis; however, their complexity may deter non-modelers.
The manuscript is well written and introduces the topic of new research but lacks in several areas; these issues must be addressed to make it interesting for the journal readers.
· Lacks graphical abstract in the manuscript.
· There are many reviews written on PBPK; what is the novelty of this review? Please mention it.
· Discuss the PBPK modeling is a feasible strategy to practice the principles of 3Rs.
· There is a need of more discussion on computational approaches such as PBPK, PK/PD, and PopPK modeling.
· The review is not revealed the search strategies, inclusion and exclusion criteria and risk of bias assessment for individual studies therefore, there is a need to add a material and methods section. What is the timeline of the review?
· Please add the conclusive remarks together with future recommendations in the separate section in Conclusion.
Author Response
Please see the attachment "Response to reviewer #2.docx".

Round 2
Reviewer 1 Report
I am satisfied with the changes made by the authors. In my opinion the paper is now publishable in Molecules as it is.
Reviewer 2 Report
· The authors have incorporated suggestions in the revised manuscript. Therefore, no issue with considering it for publication but material and methods secton shoud be after introdcution secion.